# Visual serial processing deficits explain divergences in human and VLM reasoning

## Abstract

Why do Vision Language Models (VLMs), despite strong benchmark performance, often fail on surprisingly simple visual reasoning tasks? We hypothesize that this gap reflects a deficit in visually-grounded serial processing. To test this hypothesis, we compared human and VLM performance on three domains that systematically vary serial processing load: geometric reasoning (via concept complexity), enumeration (via individuation demands), and mental rotation (via transformation difficulty). In each domain, decreased VLM accuracy was strongly correlated with increased human reaction time (used as a proxy for serial processing load). As tasks require more demanding serial processing—whether composing concepts, enumerating items, or performing mental transformations—the VLM-human performance gap widens reliably. These findings support our hypothesis that limits in serial, visually grounded reasoning form a fundamental bottleneck distinguishing current VLMs from humans.

## 1 Introduction

Large-scale vision-language models (VLMs), such as GPT-4o (Achiam et al., 2023), Claude-Sonnet 3.7 (Anthropic, 2023), and Gemini 2.5 Pro (Team et al., 2023), demonstrate impressive capabilities, often matching or surpassing human-level accuracy on complex benchmarks like image captioning and visual question answering (Fu et al., 2023). Yet, paradoxically, they also exhibit surprising deficits in tasks that appear comparatively simple, such as counting and visual search (Campbell et al., 2024b). We hypothesize that these discrepancies, and broader VLM-human divergences, reflect deficits in visually-grounded serial processing. To test this, we use tasks from cognitive science (Dehaene et al., 2006) to parametrically vary serial processing demands. We use human reaction time (RT) as a behavioral proxy for serial load: Humans adapt to complexity in visual reasoning by trading time for accuracy, using sequential attention to focus on different elements when analyzing complex scenes (Heitz, 2014; Stewart et al., 2020). VLM reasoning, by contrast, is tethered to sequential generation of text (their chain of thought), not sequential analysis of the image. We therefore predict that VLM accuracy will be inversely correlated with human RT, indicating that as tasks require more extensive serial processing by humans, the VLM-human performance gap widens. This framework enables us to localize where VLMs diverge most sharply from humans and to assess whether these divergences consistently align with operations that demand intensive serial processing.

We systematically manipulate the serial processing load in three distinct domains. First, in a geometric program induction task inspired by work modeling geometric concepts as programs (Hsu et al., 2022), we vary load by manipulating concept complexity, operationalized as the number of geometric primitives (Minimum Description Length; MDL) required to define a geometric concept (Sablé-Meyer et al., 2022). Second, for visual enumeration, drawing on work distinguishing subitizing from serial counting (Trick & Pylyshyn, 1994), serial load is varied by manipulating color distinctiveness and spatial overlap, known to increase demands on serial attention (Franconeri et al., 2013). Finally, in a mental rotation task based on Shepard & Metzler (1971), serial processing demands are modulated by varying the angular disparity between two stimuli, a factor shown to linearly increase human processing time (Cooper & Shepard, 1973). Across these diverse tasks we demonstrate that as task conditions necessitate greater serial processing the VLM-human performance gap widens reliably, providing convergent evidence for a serial processing deficit.

## 2 BACKGROUND AND RELATED WORK

### 2.1 VISION LANGUAGE MODELS

Vision Language Models (VLMs) jointly model visual and linguistic information by learning multi-modal representations that bridge the two domains (Radford et al., 2021; Alayrac et al., 2022). These models commonly pair a vision encoder with a large language model, conditioning text generation on extracted visual features (Achiam et al., 2023; Team et al., 2023; Anil et al., 2023; AI, 2024). This paradigm has driven progress on downstream tasks including image captioning, visual question answering (VQA), and complex multimodal reasoning (Liu et al., 2023; Zhu et al., 2023). Despite these advances, the mechanisms by which these systems integrate and reason about visual input remain unclear — particularly whether their strategies converge with or diverge from human visual reasoning.

### 2.2 MEASURING CORRESPONDENCES BETWEEN HUMAN VISUAL PROCESSING AND VLMS

To gauge the alignment between human and VLM visual reasoning, recent studies have benchmarked VLMs against human performance on a wide range of visual reasoning tasks from cognitive science. These comparisons have uncovered both similarities and crucial differences in their behavioral biases (Campbell et al., 2024b; Dasgupta et al., 2022). In some geometric reasoning tasks, VLMs show human-like biases (Campbell et al., 2023; 2024a), succeeding at perceptual grouping and occlusion reasoning (Allen et al., 2025; Pothiraj et al., 2025). Relative task difficulty also aligns between VLMs and humans (Xu et al., 2025), suggesting commonalities.

Yet a significant gap persists between VLM and human performance across several core cognitive tasks (Schulze Buschoff et al., 2025). VLMs struggle with spatially grounding reasoning (Tong et al., 2024; Kamath et al., 2023; Xu et al., 2025; Ramakrishnan et al., 2025), feature binding (Campbell et al., 2024b), and multi-step reasoning (Campbell et al., 2024b; Greff et al., 2020). VLM performance is near-chance on trivial tasks including determining whether circles overlap, counting, identifying circled letters (Rahmanzadehgervi et al., 2025), estimating depth, establishing visual correspondence, performing multi-view reasoning (Fu et al., 2024), and even simple visual analogies that are solvable by toddlers (Goddard et al., 2025)—all tasks where human accuracy is at ceiling.

What unites these seemingly diverse failures? This constellation of behavioral traits (Ramakrishnan et al., 2025) currently lacks a unifying explanation. Various frameworks document *when* performance falters (Schulze Buschoff et al., 2025; Ramakrishnan et al., 2025; Huang et al., 2025), but not *why*. Existing explanations range from training data constraints (Kamath et al., 2023) to deficits in core knowledge (Schulze Buschoff et al., 2025), yet no account spans the range of observed failures.

Recent work by Zhang et al. (2024) demonstrates that VLMs fail at visual deductive reasoning due to deficits in perception and hypothesis verification. We extend the investigation to a range of continuous, analog mental transformations, and suggest that a deficit in serial processing provides the missing unifying framework.

## 3 THE SERIAL PROCESSING DEFICIT HYPOTHESIS

We hypothesize that diverse VLM failures reflect a fundamental deficit in visually grounded serial processing. Three observations support this hypothesis: VLMs perform well when visual elements are well-separated but fail when they require sequential analysis (Rahmanzadehgervi et al., 2025); vision encoders contain sufficient information to solve these tasks, yet language models fail to effectively decode it (Rahmanzadehgervi et al., 2025); and tasks that cannot be easily decomposed into linguistic steps prove especially challenging (Fu et al., 2024).

We define "visual serial processing" following classic cognitive science frameworks (Treisman & Gelade, 1980; Ullman, 1984) as the iterative application of operations—such as spatial indexing, attention shifting, or analog transformation—to a visual representation. While the specific operation varies by domain, the computational bottleneck is the need to maintain and update a visual state over time.

Critically, we use Human RT not merely as a measure of difficulty, but as a proxy for this specific serial mechanism. In the domains selected, a linear increase in RT relative to task load (e.g., rotation angle) is the established behavioral signature of serial processing (Shepard & Metzler, 1971; Trick & Pylyshyn, 1994). By correlating VLM accuracy with this specific metric, we test whether models fail specifically when humans are forced to engage in serial computation.

Our contributions are threefold: First, we provide the first systematic test of serial processing as a unifying explanation for VLM failures. We demonstrate that human RT predicts model deficits, enabling forecasts of failures on novel tasks. Finally, we show that the limitation holds across model architectures and prompting strategies, indicating a structural rather than superficial gap.

## 3.1 SERIAL PROCESSING IN VLMS

Inductive biases toward serial reasoning have unlocked advances in the mathematical and coding capabilities of LLMs. The Tree of Thoughts framework (Yao et al., 2023), for instance, enables LLMs to explore, evaluate, and prune multiple reasoning pathways, tackling problems that were previously intractable. The success of such methods in language naturally motivated exploration of analogous strategies for VLMs. Initial attempts to instill serial processing in VLMs — through fine-tuning (Deitke et al., 2024) and tool use (Yang et al., 2023) — have been successful in specific task domains. Whether these initial successes will generalize across real-world tasks remains an open question. We extend this work in two ways: First, we evaluate a tool-augmented VLM across all three task domains to test if external tools offset serial processing deficits. Second, we outline training regimes aimed at instilling more generalizable visual serial reasoning.

## 4 TEST DOMAIN 1: GEOMETRIC CONCEPTS

The first domain in which we investigate a potential serial processing deficit in VLMs is geometric reasoning. A long tradition in cognitive science has examined how humans reason over geometric patterns, demonstrating that response times (RTs) increase systematically with the structural complexity of the stimulus (Just & Carpenter, 1985). A growing body of work has argued that humans interpret geometric figures by inferring the underlying generative rules that produced them, rather than merely matching surface features. This *program induction* approach has been explored in experiments that manipulate the minimum description length (MDL) of symbolic programs used to generate visual stimuli – where MDL reflects the number of discrete compositional steps required to construct the pattern from a fixed vocabulary of geometric primitives and relations (Sablé-Meyer et al., 2022). Such studies suggest that human geometric reasoning is fundamentally compositional and serial: more complex patterns require more sequential operations, resulting in longer RTs but typically preserved accuracy. Recent work introduced the *Geoclidean* domain-specific language (DSL) to generate 2D geometric stimuli by composing geometric primitives (Hsu et al., 2022). This work revealed that while humans are adept at inferring abstract geometric rules, standard transformer and CNN-based vision encoders are insensitive to this structure. However, it remains unclear whether modern VLMs, which incorporate language-based reasoning, exhibit similar deficits — particularly for complex concepts that demand greater serial processing in humans. We test this by parametrically varying the complexity of geometric concepts generated using the Geoclidean DSL, predicting that model performance will degrade with increasing complexity, even as human accuracy remains stable.

## 4.1 METHODS

We adapted the classic "oddball" paradigm from foundational studies of geometric intuition (Dehaene et al., 2006). Trials consisted of an array of six images, where five instantiate a concept and one violates it. We evaluated model and human performance on a geometric oddball task generated using the Geoclidean DSL (Hsu et al., 2022). Stimuli from 37 distinct concepts belonging to two categories — Geoclidean-Elements and Geoclidean-Constraints — were generated. (Figure 1a). These concepts spanned a range of geometric properties, including Euclidean geometry (e.g., parallelism, right angles, line bisecting), figures (e.g., squares, triangles, polygons), and constraints (tangency, midpoint relations, angle constraints). We created 100 trials for each concept, yielding a total of 3,700 trials. Human participants each completed a randomly selected subset of 50 trials, drawn from a uniformly sampled 20% subset of the model evaluation trials. These trials were sampled such that each task

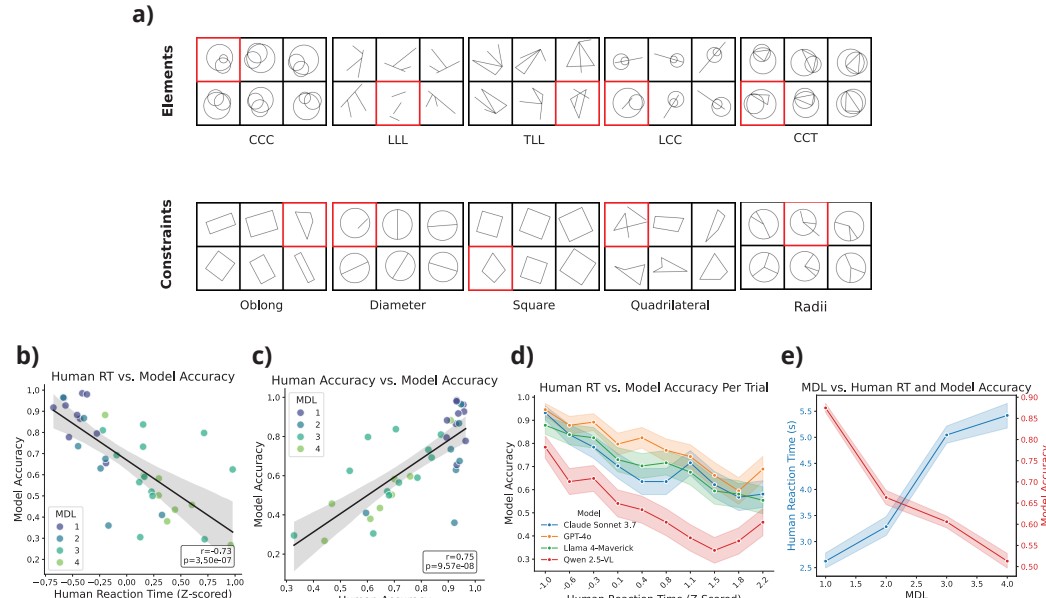

Figure 1: **Geometric Reasoning Task. a)** Example oddball detection trials from a subset of 37 geometric concepts spanning both primitive elements (left) and relational constraints (right), generated with Geoclidean DSL. **b)** Relationship between z-scored human reaction time and model accuracy; each point is one geometric concept and color denotes program complexity (Minimum Description Length, MDL). **c)** Correlation between human and model accuracy across concepts. **d)** Trial-level correlation between human reaction time (RT) and model accuracy. **e)** Human RT (blue, left axis) and model accuracy (red, right axis) as a function of MDL. Shaded regions are 95% confidence intervals.

condition was equally represented, and each trial received at least 20 independent human judgments. The oddball was constructed by removing two relational constraints from the reference concept while preserving its underlying components. Concepts ranged in program complexity from MDL 1 to 4, where Minimum Description Length (MDL) denotes the number of geometric primitives needed to generate the concept. Human participants and VLMs were tasked with identifying the position of the oddball within the array. (Appendix Figure 10). Human participants indicated their choice by clicking on the target image, and their reaction times were recorded. (Appendix Figure 9). VLMs were presented with the same array of images, each overlaid with a red index in the corner, and prompted to output the index corresponding to the oddball. (Appendix A.1.1). We evaluated four large multimodal models — GPT-4o (Achiam et al., 2023), Claude Sonnet 3.7 (Anthropic, 2023), Llama 4 Maverick (AI, 2024), Qwen2.5 VL (Bai et al., 2025) — on this task.

## 4.2 RESULTS

Human and model accuracy correlated across geometric concepts ($r = 0.75$, $p = 9.57 \times 10^{-8}$) as shown in Figure 1c, indicating shared sensitivity to difficulty. Importantly, we also observed an inverse relationship between human reaction time (RT) and model accuracy (Figure 1b). Concepts that took humans longer to solve were precisely those where models performed worst ($r = -0.73$, $p = 3.50 \times 10^{-7}$), implying that model failures are most acute on tasks requiring extended human deliberation. At the level of trials, we observed a similar inverse relationship between human RT and model accuracy (Figure 1d). We formalized this pattern by analyzing performance as a function of minimum description length (MDL). As shown in Figure 1e, human RT increased with MDL (blue), while model accuracy declined (red). This suggests that humans deploy more effort as complexity rises, whereas VLMs lack the structured inference necessary to handle deeper compositional programs.

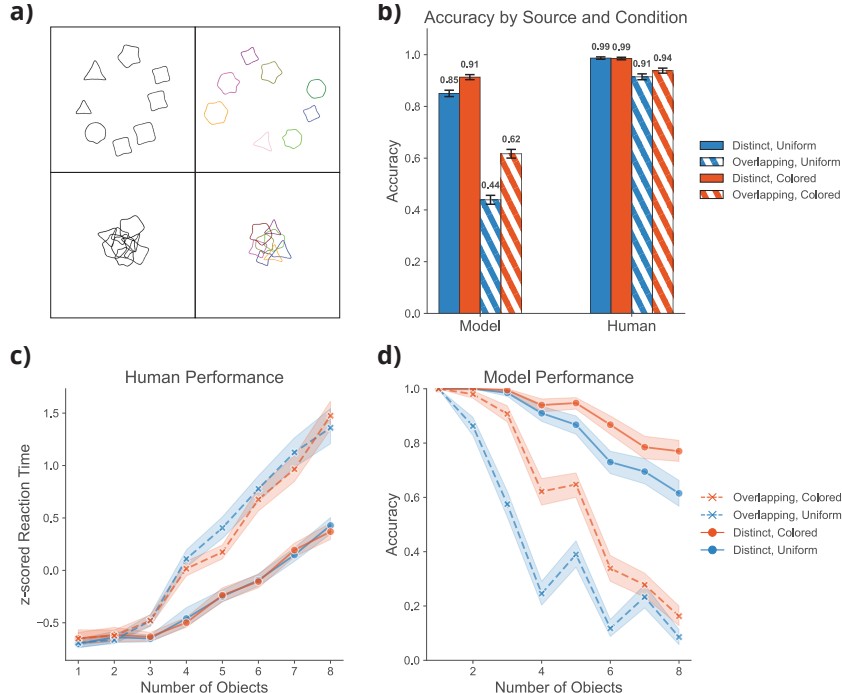

Figure 2: **Numerosity Task. a)** Example stimuli from the four experimental conditions: non-overlapping uniformly colored (top-left), non-overlapping uniquely colored (top-right), overlapping uniformly colored (bottom-left), and overlapping uniquely colored (bottom-right). **b)** Mean accuracy for humans and models across each condition, averaged across numerosities. **c)** Human z-scored reaction times as a function of numerosity across the four conditions. **d)** Model accuracy as a function of numerosity across the four conditions. Shaded regions indicate 95% confidence intervals.

These findings support our serial processing deficit hypothesis: humans accommodate complexity with slower sequential reasoning, whereas VLMs' accuracy deteriorates despite identical visual input. The following domains test whether this inverse RT–accuracy relation generalizes.

## 5 TEST DOMAIN 2: NUMERICAL ESTIMATION

Numerical estimation — such as assessing how many objects appear in a scene — is a classic serial processing task. Humans transition from near-instantaneous recognition of small sets (up to 4–6 items), a process known as subitization, to serial enumeration for larger sets (Trick & Pylyshyn, 1994; Kaufman et al., 1949). As humans enumerate larger sets or when visual discrimination becomes challenging, we deploy serial attention to identify each object — binding features to locations and maintaining distinct object representations (Mazza & Caramazza, 2015; Xu & Chun, 2009). This enumeration becomes more challenging as objects become harder to distinguish (Franconeri et al., 2013; Lavie, 2005). VLMs are known to struggle with counting (Pothiraj et al., 2025; Campbell et al., 2024b); we predict that, like humans, VLMs will perform best at low numerosities and with non-overlapping objects. However, while humans maintain accuracy in challenging conditions (Vetter et al., 2008) (e.g., overlapping objects at higher numerosities) by deploying serial attention, we predict that VLM performance will degrade rapidly. Furthermore, we predict that this degradation in performance will coincide with an increase in human reaction time. Finally, we predict that uniquely colored overlapping objects will boost VLM — but not human — performance, since color enables linguistic reference in chain-of-thought serialization.

## 5.1 METHODS

Stimuli consisted of 2-dimensional shapes generated using Hermitian splines, featuring between 3 and 5 pointed contours with randomly jittered fixed points (Figure 2a). To parametrically manipulate serial processing demands, we varied: numerosity (1-8 objects), spatial arrangement (overlapping vs. non-overlapping), and color (uniform vs. unique). This factorial design created four distinct conditions: (1) non-overlapping, uniformly colored objects; (2) non-overlapping, uniquely colored objects; (3) overlapping, uniformly colored objects; and (4) overlapping, uniquely colored objects. These manipulations were designed to vary reliance on serial attention required for object identification while controlling for the total number of objects.

In the uniform color conditions, all objects had the same hue, requiring individuation based solely on their spatial arrangement. In the unique color conditions, each object was assigned a distinct hue, providing an additional dimension for differentiation. In the non-overlapping conditions, objects were positioned with clear spatial separation, while in the overlapping conditions, objects were arranged with 60-80% intersection between adjacent shapes so boundary resolution required attention.

Each of the four conditions was instantiated across 100 trials for each numerosity level, yielding 3,200 total examples. Humans saw 50 balanced trials (see Appendix C). Sampling was stratified to ensure balanced representation across task conditions, with each trial evaluated by at least 10 independent participants. Human participants and VLMs were tasked with identifying the number of objects in the scene (Appendix Figure 12). Participants performed the task while reaction times and accuracy were recorded (Appendix Figure 11). VLMs were prompted to describe the image and report the number of objects present (Appendix A.3.1). This allowed us to directly measure the impact of increasing perceptual individuation load on task performance, and to compare the results for humans and VLMs.

## 5.2 RESULTS

Humans sustained high accuracy across conditions (Figure 2c), whereas models degraded especially with overlap. Model accuracy declined with numerosity (Figure 2d), indicating that models struggled both with high numerosities and perceptual individual loads. These same conditions took humans longer to solve ($r = -0.97$, $p = 8.22 \times 10^{-5}$) and we observed an inverse relationship between human RT and model performance. Performance was also consistently impaired on overlapping trials, with models showing substantially lower accuracy ($t = -8.43$, $p = 7.9 \times 10^{-10}$), and humans requiring more time ($t = 3.45$, $p = 0.005$) and exhibiting reduced accuracy ($t = -2.43$, $p = 0.023$).

Finally, we observed that model performance was significantly better in the colored overlapping condition compared with the uniform overlapping condition ($t = 5.90$, $p = 8.1 \times 10^{-7}$; Figure 2b). This model-specific improvement was not accompanied by a corresponding decrease in human reaction time ($t = -1.17$, $p = 0.28$; Figure 2c) or accuracy ($t = 2.06$, $p = 0.078$), suggesting that color aided model performance not by reducing perceptual difficulty, but by facilitating object individuation. This supports our hypothesis that VLM serialization is improved when objects can be uniquely referenced by their color in context, a capability that humans do not benefit from.

## 6 TEST DOMAIN 3: MENTAL ROTATION

Mental rotation provides a final test case where the sequential nature of the task cannot be effectively substituted with linguistic reasoning. Unlike symbolic pattern recognition, mental rotation requires the manipulation of spatial representations through continuous transformations—operations that are difficult to express in discrete, language-based tokens. A standard mental rotation task presents participants with two images which are either two distinct objects or one object shown from different angles. Decades of cognitive science research demonstrates that humans solve these tasks by mentally rotating one object to align with another, an analog process captured by the robust linear relationship between angular disparity and reaction time (Shepard & Metzler, 1971; Cooper & Shepard, 1973). Functional imaging confirms that this process involves incremental spatial adjustments, engaging parietal regions known to support visuospatial transformation (Zacks et al., 1999; Just & Carpenter, 1985). These transformations rely on what Kosslyn (1994) termed "depictive representations" – mental images that preserve geometric structure and spatial relations. Because such analog operations

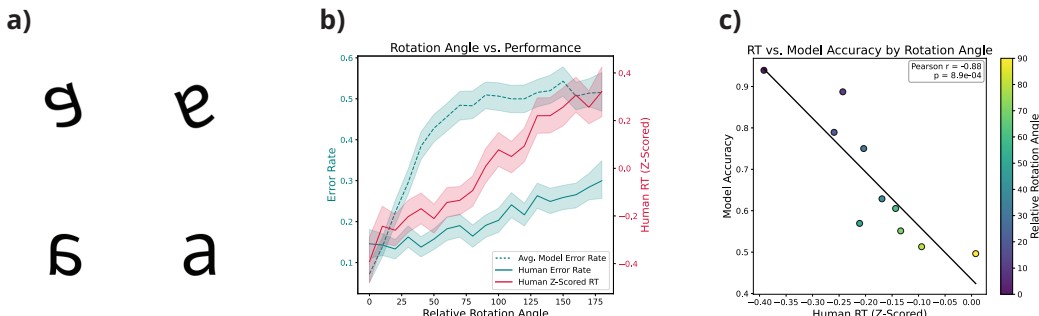

Figure 3: **Mental Rotation Task. a)** Example stimuli from the mental rotation task. Participants and models judged whether a rotated letter was the same or a mirror-reversed version of a reference. **b)** Human error rate (solid line), model error rate (dashed line), and z-scored human reaction time (right axis) plotted against relative rotation angle. Shaded regions denote 95% confidence intervals. **c)** Relationship between z-scored human reaction time and model accuracy across rotation angles ($\leq 90°$). Each point corresponds to a single rotation angle; colors indicate relative angular disparity.

are difficult to serialize into linguistic prompts, mental rotation offers a particularly stringent test of whether VLMs can support visual reasoning that is not scaffolded by language.

## 6.1 METHODS

We adapted a classic mental rotation task from prior cognitive science paradigms (Shepard & Metzler, 1971; Cooper & Shepard, 1973). Stimuli consisted of character-like shapes that were either identical or mirror-reversed versions of a reference, displayed at various rotation angles (Figure 3a). Each trial presented a pair of shapes, and the task was to determine whether they matched or were mirrored. (Appendix Figure 14).

Rotation angles varied systematically from 0° to 360° in 10° increments, yielding 36 distinct orientations. For each angle and letter stimulus (26 total), we generated four trial types crossing two factors: whether the pair was the same or mirror-reversed, and whether the first image was mirrored or not. This design resulted in $36 \times 4 \times 26 = 3744$ total trials. Order was randomized and balanced across conditions. Humans completed 50 balanced trials each (Appendix Figs. 13,14), reporting "same" vs. "different" with RT recorded. Each trial was judged by at least 10 participants. VLMs received indexed images and prompts to classify (Appendix A.2.1). Model performance was evaluated as overall accuracy across each numerosity and load condition.

## 6.2 RESULTS

Humans reproduced the classic linear RT–angle effect (Fig. 3b): larger angle disparities yielded longer RTs, consistent with analog transformation (Just & Carpenter, 1985; Zacks et al., 1999).

As predicted, VLM performance exhibited an inverse relationship. As rotation angle increased, model accuracy declined, with particularly sharp drops in accuracy beyond 75°, consistent with prior findings on spatial reasoning limitations in vision-language systems (Stogiannidis et al., 2025; Chen et al., 2025). This decline is reflected in the average model error rate (dashed line, Figure 3b), which rose substantially across angle bins.

Human accuracy also declined with rotation angle, but more modestly. While participants made more errors at higher angles, their performance degraded far less steeply than that of the models, indicating partial robustness to increased serial demands. This distinction reinforces the idea that humans can compensate for increasing task difficulty by allocating more cognitive resources—reflected in longer RTs—whereas VLMs currently lack such dynamic inference capabilities.

Critically, we observed a strong negative correlation between human RT and model accuracy across rotation angles (Figure 3c; $r = -0.88$, $p = 8.9 \times 10^{-4}$). Trials that demanded longer processing times from humans were precisely those where VLMs performed worst. This inverse relationship

supports the serial processing deficit hypothesis: the tasks that elicit deeper, stepwise transformations in humans are the ones that reveal structural limitations in current VLMs.

Together, these findings suggest that mental rotation engages a form of non-linguistic serial reasoning that vision-language models do not replicate. Despite identical visual input, models falter where humans invoke continuous internal transformations.

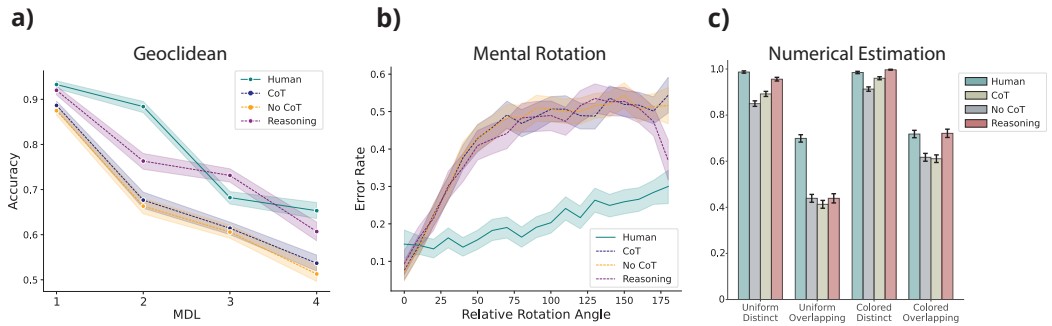

Figure 4: **Results for Augmented VLMs.** **a)** Accuracy as a function of MDL for humans and different model conditions. **b)** Error rate as a function of relative rotation angle for humans (green line) and models under different augmentation paradigms. **c)** Mean accuracy for humans and models across each condition, averaged across numerosities. Shaded regions in **a)** and **b)** represent 95% confidence intervals.

## 7 CAUSAL EVIDENCE FOR A SERIAL PROCESSING DEFICIT

One way to causally test the hypothesis that VLMs lack serial processing capabilities is to augment VLMs with a form of serial processing, and test whether this enables them to solve more complex visual tasks. To investigate this, we tested models augmented with three forms of serial processing: Chain-of-thought (CoT), reasoning training, and tool use (e.g., the ability to crop and rotate images).

We do not expect these forms of serial processing to be effective in all tasks. Improvements from CoT and reasoning training should be limited to cases where individual objects can be uniquely referenced in the models' CoT. Our numerosity estimation task provides an ideal setting to test this prediction. In this task, when objects are both non-overlapping and distinctly colored, textual CoT can effectively be used to describe the objects one at a time (based on color), whereas textual CoT should be less effective when objects are overlapping or uniform in color. Consistent with this, we see that this condition (colored distinct) is the one where CoT ($t = 7.79$, $p < 0.005$) and reasoning training ($t = 13.81$, $p < 0.005$) help the most. Reasoning training results in nearly perfect performance in this condition (99.7% accuracy; Figure 4), but does not improve performance significantly in the overlapping conditions ($p > 0.5$).

Similarly, improvements from tool use should be limited to cases where the task can be solved by a simple visual transformation. As detailed in Appendix 2, we evaluated tool use across all three domains with the GPT-o3 model. This results in near-perfect accuracy on our rotation task (Table 2, $p < 0.05$) and perfect performance on conditions in which the objects are spatially separated (uniform distinct and colored distinct) in our numerosity task (100% accuracy, $p < 0.005$). However, tool use does not improve on baseline performance for either of the overlapping conditions (Table 2, $p > 0.1$).

Two major conclusions can be drawn from these results. First, they show that augmenting VLMs with serial processing enables them to solve visually complex tasks, thus providing causal evidence for the serial processing deficit hypothesis. Second, these results serve as a diagnostic: they highlight that existing methods for serial processing in VLMs (textual CoT and tool use) are limited to specific settings (e.g., where linguistic grounding is possible or objects are spatially separable). This points to the need for developing intrinsically visual forms of serial processing to help VLMs deal with a broader range of visually complex tasks.

## 8 DISCUSSION

Our experiments show that VLMs consistently fail where tasks demand visually-grounded serial processing, revealing behavioral limitations that parallel specific constraints in human cognition. Across geometric reasoning, enumeration, and mental rotation, VLM accuracy fell precisely where human reaction time (RT) rose—supporting the serial processing deficit hypothesis. The cross-domain consistency suggests a general computational constraint, not a domain-specific weakness. This framework moves beyond the "binding problem" by identifying the underlying mechanism: VLMs lack the serial processing required to bind features (Treisman & Gelade, 1980) or update spatial states (Shepard & Metzler, 1971), explaining failures across both static and dynamic tasks.

In geometric reasoning, increasing program complexity (MDL) led to longer human RTs but declining model accuracy. Earlier work using similar tasks argued that human performance implies symbolic program induction, a capability that neural network models lack (Sablé-Meyer et al., 2022). Contrary to this finding, we show that VLMs *do* correlate with both human behavior and measures of task complexity. It is an open question whether to interpret this as evidence that VLMs perform program induction, or that the human behavioral measures do not reflect program induction. However, our results indicate that VLMs appear unable to flexibly compose visual operations in ways that scale with program complexity.

In numerical estimation, models displayed human-like subitizing for small sets but degraded rapidly under individuation load. Humans preserved accuracy by deploying serial attention (Mazza & Caramazza, 2015; Xu & Chun, 2009) at the cost of increased processing time. We also found a novel asymmetry: VLMs benefit from unique coloring in overlapping object conditions, while humans show no comparable advantage. This contrasts with some findings, such as those by Rahmanzadehgervi et al. (2025), who reported that coloring conditions in numerosity tasks made no significant difference for most of the VLMs tested. Here, we show that when objects overlap, unique coloring provides a distinct and significant advantage to VLMs. This advantage likely arises because distinct colors allow VLMs to individually reference and effectively count objects, thereby enabling them to linguistically ground features and perform feature binding in a way that is not possible in uniformly colored conditions.

In mental rotation, humans showed a linear relationship between rotation angle and reaction time (Shepard & Metzler, 1971; Cooper & Shepard, 1973) (reflecting an incremental transformation process), while models showed steep accuracy declines beyond moderate angles. This suggests a limitation in performing the continuous transformations that humans execute through depictive representations (Kosslyn, 1994). Importantly, we are not claiming that models are incapable of serial processing — indeed, autoregressive generation implicitly implements serial processing. However, we distinguish between *linguistic serialization* (generating sequential text tokens) and *visual serialization* (applying sequential operations to a visual representation). While VLMs possess the former, our results suggest they lack the latter. This interpretation is supported by established psychophysics literature. When human exposure time is restricted—forcing reliance on parallel processing—human performance profiles collapse to match those of VLMs. For example, restricting time eliminates the ability to count large sets (Mandler & Shebo, 1982), creates binding errors (Treisman & Gelade, 1980), and prevents accurate mental rotation (Hertzog et al., 1993). That "time-restricted humans" behave like VLMs strongly suggests VLMs behave more similarly to parallel-only visual reasoners.

These results offer an explanatory account of why models so adept at many tasks struggle with activities as simple as counting. They also point toward concrete strategies for improving VLM performance by targeting improvements in visual serial processing.

### 8.1 ARCHITECTURAL IMPROVEMENTS

Recent VLM augmentations — such as V* (Yang et al., 2023) and tool use — are intended to mimic how humans process scenes. Humans shift their attention across regions rather than processing entire images at once, using high-resolution central vision to examine areas in detail (Stewart et al., 2020). Our work shows two fundamental limitations of these current augmentations. First, linguistic Chain-of-Thought (via prompting and reasoning fine-tuning) fails when input cannot be grounded in language. Second, tool use succeeds for explicit transformations, but not for general serial reasoning.

A more promising direction is Visually Grounded Reinforcement Learning (RL). Methods such as ViGoRL, GRIT, UniVG-R1, and Ground-R1 (Sarch et al., 2025; Wang et al., 2025; Li et al., 2025; Zhao et al., 2025) train policies that generate reasoning traces anchored to coordinates or bounding boxes, computationally analogous to the human saccade-and-fixate cycle. This forces models away from holistic strategies toward structured, pseudo-serial scene analysis, while also improving interpretability by making attentional paths explicit. In our Appendix, we analyze internal attention maps as a first step towards mechanistically defining serial processing failures (Appendix E). Our results reveal that current models fail to serially individuate overlapping objects, treating them instead as a diffuse group.

Post-training methods inspired by serial vision also show domain-specific promise. Molmo (Deitke et al., 2024) improves counting by learning to point to objects sequentially. Similar fine-tuning could extend to other domains. More generally, deficits we observed — stemming from interferences between visual elements (Campbell et al., 2024b) — could be reduced by causal masking around fixation points (as in Molmo), paralleling the human distinction between foveal and peripheral vision (Stewart et al., 2020). Recent work has also explored allowing VLMs generate and modify visual content via integrated diffusion architectures, aiming to replicate the reasoning benefits of human mental imagery (Wiedemer et al., 2025). Such controlled, sequential attention offers a path toward more human-like serial visual processing.

## 8.2 LIMITATIONS & FUTURE DIRECTIONS

Our analysis has two main limitations. First, we evaluated only a subset of competitive VLMs; broader sampling may reveal model-specific variation. Second, we focused on three domains designed to manipulate serial load. Extending to additional paradigms — such as visual search and spatial reasoning (Gilden et al., 2010; Reimer & Schubert, 2019) — would test the generality of our claims and help disentangle serial from purely spatial deficits.

## 8.3 CONCLUSION

Our work provides a unifying explanation for the discrepancy between VLMs' impressive benchmark performance and their difficulty with seemingly simple visual reasoning tasks. The serial processing deficit hypothesis offers both a theoretical framework for understanding current model limitations and a roadmap for future architectural innovations. By pinpointing operations where VLMs diverge from humans, we advance understanding of the computational principles behind visual intelligence. Addressing serial processing explicitly — integrating parallel pattern recognition with structured sequential reasoning — may be essential for the next generation of VLMs to achieve human-like flexibility.

## 9 ETHICS STATEMENT

We conducted our human behavioral experiments under an IRB-approved protocol. Further information, such as about the potential risks of behavioral data collection and details about compensation, can be found in the Appendix.

We address potential societal impacts of our work in the Discussion. Our experiments do not pose a high risk for misuse since we do not release new models or use any proprietary assets.

## 10 REPRODUCIBILITY STATEMENT

All experimental details and methods for testing models are described in the main text and Appendix. The code base necessary to generate experimental stimuli, run model experiments, and human behavioral experiments will be open-sourced upon publication. We will include our full datasets and instructions to faithfully reproduce our work.

We include screenshots of instructions provided to participants in the Appendix, as well as details on compute needed to run model experiments. Our work does not involve any novel theory that would require formal justification. Finally, we report statistical significance for all results where appropriate in the main text and Appendix.

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

# A APPENDIX: PROMPTS FOR VISION-LANGUAGE MODEL EXPERIMENTS

## A.1 GEOCLIDEAN

### A.1.1 BASELINE PROMPT

```
Which of the 6 shapes is different from the others?

Return only the number of the odd one out in square brackets (e.g., [3]) without any
   additional reasoning or justification.
```

### A.1.2 CHAIN-OF-THOUGHT PROMPT

```
Which of the 6 shapes is different from the others?

Analyze the relations between the parts of each stimulus to identify the stimulus that is
   different from the rest. Return your reasoning followed by the integer label of the odd one
    out enclosed in square brackets (e.g., [3]).
```

## A.2 ROTATION

### A.2.1 BASELINE PROMPT

```
You will be shown two objects that are rotated at different angles. Your task is to determine
    whether these objects are:

A) The **SAME** object shown at different rotations
B) **DIFFERENT** objects that are mirror images of each other (these mirror images may also
   be rotated at different angles)

Even when rotated, identical objects will have the same features in the same arrangement.
   Mirror images will have reversed features that cannot be matched by rotation alone.

## Response Format:
[1] - The objects are identical but rotated differently
[0] - The objects are mirror images of each other

Please just include your final answer in the format [1] or [0] without any additional
   reasoning or justification.
```

### A.2.2 CHAIN-OF-THOUGHT PROMPT

```
You will be shown two objects that are rotated at different angles. Your task is to determine
    whether these objects are:

A) The **SAME** object shown at different rotations
B) **DIFFERENT** objects that are mirror images of each other (these mirror images may also
   be rotated at different angles)

Even when rotated, identical objects will have the same features in the same arrangement.
   Mirror images will have reversed features that cannot be matched by rotation alone.

## Response Format:
[1] - The objects are identical but rotated differently
[0] - The objects are mirror images of each other

Please include your reasoning followed by your final answer in the format [1] or [0].
```

## A.3 NUMEROSITY

### A.3.1 BASELINE PROMPT

```
The following images contains 1 or more objects that may or may not be overlapping.

Count the total number of objects in the scene, and return your final answer as an integer
   enclosed in square brackets (e.g., [10]).
```

### A.3.2 CHAIN-OF-THOUGHT PROMPT

The following images contain 1 or more objects that may or may not be overlapping.

Count the total number of objects in the scene one at a time. Once you have described all
    objects, return your final answer as an integer enclosed in square brackets (e.g., [10]).

# B APPENDIX: INDIVIDUAL MODEL RESULTS

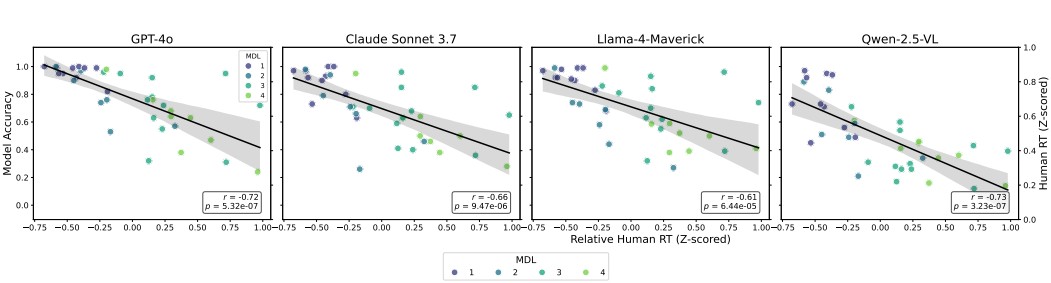

Figure 5: **Human RT vs. Model Accuracy by Model.** Z-scored human reaction time and model accuracy plotted across geometric concepts, shown separately for each VLM. Each point corresponds to one concept, colored by program complexity (MDL). Black lines show linear regression fits with 95% confidence intervals.

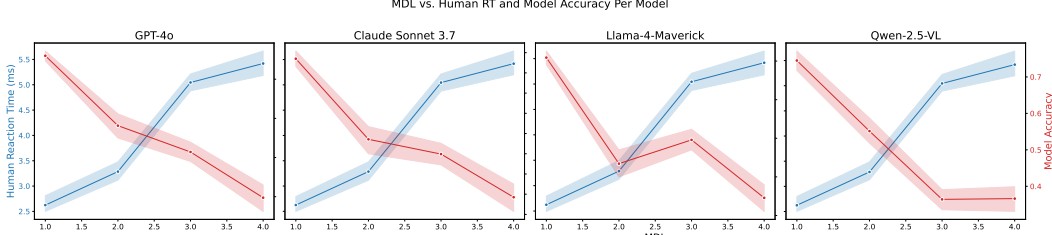

Figure 6: **MDL vs. Human RT and Model Accuracy by Model.** Human reaction time (blue, left axis) and model accuracy (red, right axis) plotted as a function of MDL, shown separately for each VLM. Shaded regions indicate 95% confidence intervals.

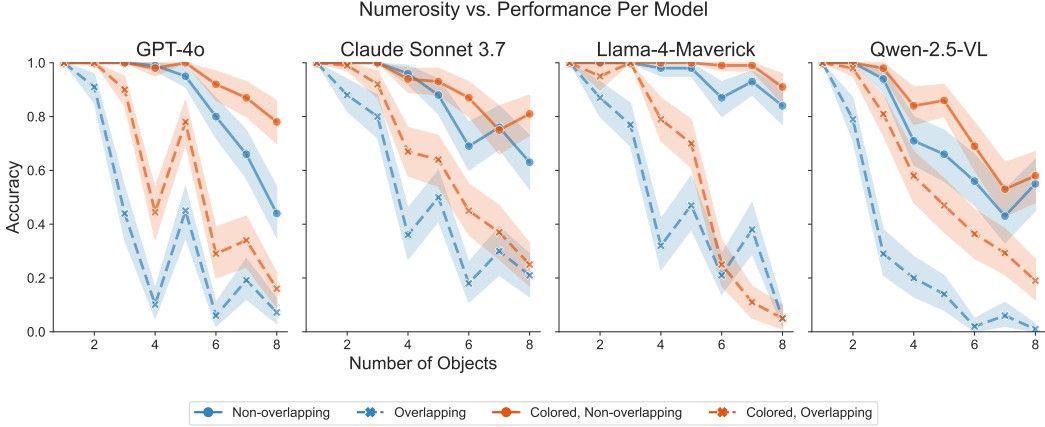

Figure 7: **Numerosity vs. Performance by Model.** Model accuracy in the overlapping condition (dashed), distinct condition (solid), colored condition (orange), and uniform condition (blue) plotted as a function of the number of objects in the scene, shown separately for each VLM. Shaded regions indicate 95% confidence intervals.

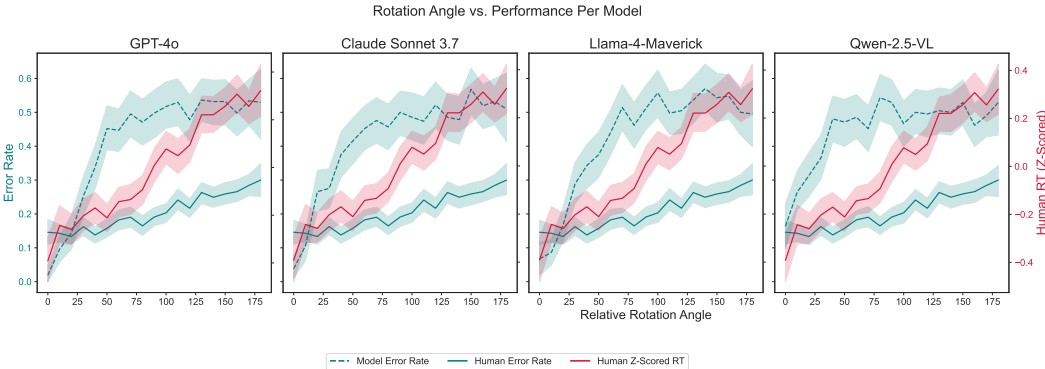

Figure 8: **Rotation Angle vs. Performance by Model.** Model error rate (dashed), human error rate (solid), and z-scored human reaction time (right axis) plotted as a function of relative rotation angle, shown separately for each VLM. Shaded regions indicate 95% confidence intervals.

## C  APPENDIX: HUMAN BEHAVIORAL EXPERIMENTS

Human behavioral data for the geometric complexity, numerical estimation, and mental rotation tasks were collected via Prolific under an IRB-approved protocol. To maintain double-blind review standards, we do not include the consent form. In the geometric complexity task, participants selected the oddball from an array of geometric objects. In the numerical estimation task, they counted the total number of objects presented. In the rotation task, they determined whether a pair of objects were mirrored versions of each other. Participants were compensated at an approximate rate of $12 per hour.

### C.1  GEOCLIDEAN

Which of the following geometric shapes is the outlier? Once you press the cross, click on the outlier as quickly as you can.

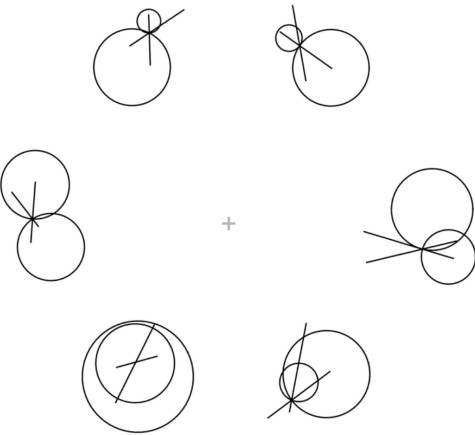

Figure 9: **Example of Geoclidean Task.**

Thank you for participating in our study!

### Task Instructions

In this study, you will be shown six geometric shapes in each trial. Five shapes will be similar, while one shape will be different - this is the "outlier". Your task is to find and click on the outlier as quickly as possible.

### Important Notes

Please note that while the shapes may appear in different sizes and orientations, these differences do not matter. Each shape is made up of the same basic parts, but what's important is how these parts relate to one another. The five reference shapes will share the same relationships amongst parts, while the outlier shape will have these same parts joined together differently. Focus only on identifying this outlier. Even if you are unsure, you must make a choice in each trial.

### How to Respond

Speed is essential in this task. You can take your time to prepare before each trial begins, but once you click the central cross to start, you should identify and click the outlier shape as quickly as you can. After each trial you will receive feedback.

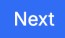

Figure 10: **Instructions for Geoclidean Task.**

## C.2 NUMEROSITY

How many objects do you see in this image? Once you start the trial, count the objects and enter your answer as quickly and accurately as possible. (Press Enter or click the cross button to start the trial).

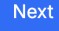

Submit
Answer

Figure 11: **Example of Numerosity Task.**

Thank you for participating in our study!

## Task Instructions

In this study, you will be shown a series of images containing objects. Your task is to count how many objects are present in each image as quickly and accurately as possible.

## How to Respond

Both speed and accuracy are essential for this task. You will receive a reward based on the number of correct answers you provide.

Feel free to take your time to prepare before each trial. Once you're ready, click the central cross or press the Enter key to start. Then, count the objects as quickly and accurately as possible and enter your answer in the number field. After each trial you will receive feedback.

Next

Figure 12: **Instructions for Numerosity Task.**

## C.3 ROTATION

Are these two images the same letter (rotated versions) or mirror images of each other? Click the cross to begin, then select either "Same" or "Different" as quickly as you can.

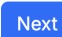

Figure 13: **Example of Rotation Task.**

Thank you for participating in our study!

## Task Instructions

In this study, you will be shown pairs of letter stimuli. Your task is to determine whether the two letters are the same (only rotated versions of each other) or different (mirror images of each other).

## How to Respond

Both speed and accuracy are essential for this task. You will receive a reward based on the number of correct answers you provide.

Feel free to take your time to prepare before each trial. Once you're ready, click the central cross to start.
After you click the central cross, you should identify whether the two letters are the same or different as quickly as you can by clicking the appropriate button. After each trial you will receive feedback.

Next

Figure 14: **Instructions for Rotation Task.**

# D  APPENDIX: CAUSAL EVIDENCE FOR A SERIAL PROCESSING DEFICIT

## D.1  CHAIN-OF-THOUGHT AND REASONING

Chain-of-Thought experiments were conducted with GPT-4o, Claude Sonnet 3.7, Llama 4 Maverick, and Qwen2.5 VL, each prompted with the Chain-of-Thought templates described in Appendix A. Reasoning experiments were carried out with GPT-o3, Gemini 2.5 Pro, and Claude Opus 4, also using the Chain-of-Thought prompts. All three experimental paradigms were completed in full under the conditions specified in the main paper. The full set of results for these models, compared against humans, are reported in Table 1.

| Model | Geoclidean (CI) | Numerosity (CI) | - | - | - | Rotation (CI) |
|-------|-----------------|-----------------|---|---|---|---------------|
| | | uniform_distinct | uniform_overlapping | colored_distinct | colored_overlapping | |
| **Human** | 78.2 (77.5–78.9) | 98.7 (98.1–99.1)% | 69.9 (68.3–71.5)% | 98.5 (97.9–98.9)% | 71.8 (70.2–73.4)% | 79.6 (78.9–80.3)% |
| **NoCoT** | 66.9 (66.2–67.7)% | 85.0 (83.8–86.2)% | 43.9 (42.2–45.6)% | 91.3 (90.3–92.2)% | 61.7 (60.0–63.4)% | 56.7 (55.9–57.5)% |
| **CoT** | 68.5 (67.7–69.2)% | 89.2 (88.1–90.3)% | 41.3 (39.6–43.0)% | 96.0 (95.3–96.6)% | 61.1 (59.4–62.7)% | 56.9 (56.1–57.8)% |
| **Reasoning** | 76.2 (75.4–77.0)% | 95.6 (94.7–96.3)% | 43.9 (41.9–45.9)% | 99.7 (99.5–99.9)% | 72.1 (70.3–73.9)% | 58.1 (57.1–59.0)% |

Table 1: Mean accuracy (95% CI) for humans and all model classes (NoCoT (Baseline), CoT, and Reasoning) across the full evaluation set, spanning Geoclidean, Numerosity (four conditions), and Rotation tasks.

## D.2  TOOL-USE

Tool-use experiments were conducted through the ChatGPT interface with the GPT-o3 model. This interface provides the model with advanced, agentic visual analysis tools, including object extraction, cropping/zooming, and code execution for image analysis. Across three experiments, we performed a total of 400 trials, focusing on challenging task conditions: 100 trials on Geoclidean tasks (20 for each of five tasks, including two with MDL 3 and three with MDL 4), 100 trials on Rotation tasks (angles from 90° to 270° in 30° increments), and 200 trials on Numerosity tasks ($n = 8$ objects, 50 trials for each of the four conditions). The model was prompted with the Chain-of-Thought prompts, as detailed in Appendix A. Results on this restricted set of 400 challenging trials are shown in Table 2.

| Model | Geoclidean (CI) | Numerosity (CI) | - | - | - | Rotation (CI) |
|-------|-----------------|-----------------|---|---|---|---------------|
| | | uniform_distinct | uniform_overlapping | colored_distinct | colored_overlapping | |
| **Human** | 58.0 (55.7–60.2)% | 96.9% (92.3–98.8%) | 73.5% (66.3–79.6%) | 93.8% (88.7–96.7%) | 73.9% (65.4–81.0%) | 75.2 (71.5–78.6)% |
| **NoCoT** | 45.5 (40.7–50.4)% | 58.5% (51.6–65.1%) | 9.5% (6.2–14.4%) | 75.0% (68.6–80.5%) | 14.5% (10.3–20.0%) | 47.8 (43.1–52.5)% |
| **CoT** | 43.5 (38.7–48.4)% | 70.9% (64.2–76.7%) | 6.0% (3.5–10.2%) | 78.4% (72.2–83.5%) | 18.1% (13.4–24.0%) | 47.3 (42.6–52.2)% |
| **Reasoning** | 49.3 (43.7–55.0)% | 79.2% (72.0–84.9%) | 8.7% (5.1–14.3%) | 99.3% (96.3–99.9%) | 29.3% (22.6–37.1%) | 48.4 (43.0–53.9)% |
| **Tool Use** | 66.0 (56.3–74.5)% | 100.0% (92.9–100.0%) | 10.0% (4.3–21.4%) | 100.0% (92.9–100.0%) | 28.0% (17.5–41.7%) | 88.8 (81.0–93.6)% |

Table 2: Mean accuracy (95% CI) for humans and all model classes (NoCoT (Baseline), CoT, Reasoning, and Tool Use) evaluated on the same 400 challenging trials spanning Geoclidean, Numerosity (four conditions), and Rotation tasks.

# E APPENDIX: INTERNAL ATTENTION ANALYSIS

To investigate the mechanistic basis of the serial processing deficit, we analyzed the internal attention weights of Qwen2.5-VL-7B and LLaVA-v1.6-Mistral-7b on the Numerosity task. We hypothesized that the performance deficit in overlapping conditions stems from a failure to serially individuate objects during encoding.

## E.1 METHOD

We generated attention maps over the image tokens for each generated text token. We evaluated the **object-centricity** of attention by calculating the Pearson correlation of attention fixations between different objects in the scene. For each trial, we computed an "attention time-series" for each object (average attention within its mask per token). We then averaged the pairwise correlations between objects. A high correlation indicates the model attends to objects in unison (diffuse/grouped attention), while a low correlation indicates independent, selective attention (individuation).

## E.2 RESULTS

As shown in Table 3, both models exhibited significantly higher attention correlations in the overlapping condition compared to the non-overlapping condition (Qwen: $t = 14.06$, $p < 0.001$; LLaVA: $t = 2.74$, $p = 0.006$). This indicates that spatial overlap causes attention to become synchronized and diffuse, preventing the serial scanning required for counting. LLaVA also showed a significant reduction in correlation for colored vs. uniform objects ($t = -18.57$, $p < 0.001$), supporting the hypothesis that color aids linguistic grounding of attention.

| Model | Condition Comparison | Mean Correlation (Cond A vs B) | t-stat | p-value |
|---|---|---|---|---|
| **Qwen2.5-VL** | Overlap vs. Non-Overlap | 0.841 vs. 0.622 | 14.06 | $1.34 \times 10^{-34}$ |
| | Colored vs. Uniform | 0.745 vs. 0.718 | 1.44 | 0.15 |
| **LLaVA-v1.6** | Overlap vs. Non-Overlap | 0.852 vs. 0.834 | 2.74 | 0.006 |
| | Colored vs. Uniform | 0.799 vs. 0.888 | -18.57 | $4.09 \times 10^{-52}$ |

Table 3: Attention correlation analysis. Higher correlations indicate a failure to individuate objects (diffuse attention). Overlap significantly degrades individuation across models.

# F APPENDIX: COMPUTATIONAL RESOURCES

All model queries were conducted via API (e.g., Together), including open-weight models such as Llama. No specialized hardware was required. Dataset generation and preprocessing were lightweight and completed on standard CPU machines.

