# OpenReview forum: "Visual serial processing deficits explain divergences in human and VLM reasoning"
_ICLR.cc/2026/Conference — ICLR 2026 Conference Desk Rejected Submission_

### Official Review · Reviewer_kB3k · 2025-10-16

**Soundness:** 3
**Presentation:** 3
**Contribution:** 2
**Rating:** 2
**Confidence:** 3

**Summary:**

This paper proposes an hypothesis to explain some limitations of Visual Language Models: This is due to a deficit of serial processing on the visual side. In order to assess that hypothesis, several tests are driven on several SOTA VLM, related to geometric reasoning, enumeration and mental rotation. It is indeed observed that errors are more important when the serial aspect of that task was increasing. Note that this increasing serial aspect is measured by an increasing processing time. It is concluded that these observations support the hypothesis.

**Strengths:**

The hypothesis is simple and clearly stated. The experimental protocols are carefully described.

**Weaknesses:**

Globally, I found the contribution relatively weak.
-the unique and central hypothesis of this paper is just stated, with no justification or intuition or elaborated development. It is just given.
- the main contribution is about the assessment of a rather simple and obvious hypothesis. I agree that it is important  to precisely evaluate it but the impact of the paper is not so important.
- the fact that the level of complexity of the serial task is directly and uniquely measured by the processing time should be discussed more. I am not convinced it is always true
-the conclusion corresponds to a generality, whereas the test was only carried out for three kinds of tasks and some VLM.
-beyond this conclusion, it could have been interesting to discuss more implications or possible adaptations of VLM.

**Questions:**

Are you certain that the level of complexity is directly measured by the processing time ?

---

> ### Author Response · Authors · 2025-11-22
>
> We thank Reviewer kB3k for their review. We believe there may be fundamental misunderstandings regarding the paper's motivation, the cognitive science literature grounding our methodology, and our contributions.
> ## Justification of the Hypothesis
>
> The reviewer states that the hypothesis is "just stated, with no justification or intuition."
>
> - The hypothesis is derived in Section 3 from three limitations in prior work: (1) failures in sequential visual analysis (Rahmanzadehgervi et al., 2025), (2) failure to decode encoder information (ibid), and (3) difficulty with non-linguistic tasks (Fu et al., 2024).
> - Intuition: The intuition is that while VLMs are linguistically serial (generating text token-by-token), they are visually parallel (encoding the image in a single pass). Our hypothesis provides the missing unifying framework to explain why models that excel at complex semantics fail at "simple" tasks like counting: they lack the iterative visual sub-routines (scanning, rotating, tracing) required to solve them.
>
> We have clarified these points in the revised paper, making it apparent where the hypothesis comes from.
> ## Novelty & Impact
>
> The reviewer argues the hypothesis is "simple and obvious" and the impact "not so important."
> - Scientific value: A simple, parsimonious hypothesis explaining diverse failures is valuable. The field currently lacks a predictive account for when and why these models fail.
> - Novel contribution: Our contribution is not just stating the hypothesis, but proving it holds across three distinct cognitive domains and, crucially, establishing a link between human psychophysics and VLM performance. To our knowledge, no prior work has demonstrated that human Reaction Time (RT) serves as a continuous predictor of VLM failure ($r > -0.9$ in our experiments). This moves the field from benchmarking to behavioral modeling.
>
> We have clearly identified these contributions in the revised paper.
> ## Validity of RT as a Measure of Seriality
>
> The reviewer asks if the level of complexity is directly measured by the processing time.
>
> RT in general reflects difficulty. However, in human vision, the mechanism for handling increased difficulty in these specific domains is serial processing. Specifically, it is not just that the tasks are hard, but that human RT scales linearly with the parameter of interest (e.g., number of items, rotation angle). In cognitive science, this linear slope is the signature of serial processing (Shepard & Metzler, 1971).
>
> We do not claim RT always equals seriality. We claim that in tasks where human RT exhibits this linear serial signature, VLMs fail. This specific validity is what makes our findings robust.
>
> We have revised Section 3 to explicitly cite the foundational literature (Shepard & Metzler, 1971; Trick & Pylyshyn, 1994) that grounds this methodology. We clarify that we do not claim RT always equals seriality, but that the linear RT signature allows us to strictly define when humans rely on serial reasoning strategies.
> ## Generality (Only 3 Tasks)
>
> The reviewer notes the test was only carried out for three kinds of tasks.
> - We deliberately chose these three tasks because they are the cornerstones of non-linguistic cognition and widely studied in cognitive psychology. These represent three distinct types of serial processing: Spatial Transformation (Mental Rotation), Visual Attention/Individuation (Numerical Estimation), and Geometric Program Induction.
> - Finding a consistent inverse correlation between human RT and VLM accuracy across these diverse mechanisms provides strong convergent evidence for a general underlying deficit in the architecture, rather than a task-specific failure.
> We have explained this motivation in the revised paper.
> ## Implications and Adaptations
>
> The reviewer suggests discussing implications. We explicitly address this in Section 8.1 (Architectural Improvements), but we have expanded this analysis with new experiments in response to your feedback:
> - Internal Mechanism Analysis: We conducted a new attention analysis on open-source models (Qwen2.5, LLaVA). We found that in conditions where humans use serial attention (overlapping objects), VLM attention becomes diffuse and highly correlated across objects. This confirms the implication that current architectures fail to individuate objects during the encoding phase. For a more detailed account, please see our response to reviewer Ddut.
> - Concrete Path Forward: We identify two concrete adaptations for future VLMs: (1) Visually Grounded Reinforcement Learning (training models to generate "saccades" or bounding boxes sequentially), and (2) Recurrent Vision Encoders that allow the visual state to be updated iteratively, mirroring human working memory.
>
> We believe that clarifying the rigorous cognitive science foundations of our methodology demonstrates that this contribution is neither weak nor obvious, but a necessary step in understanding the fundamental limits of current architectures.

---

### Official Review · Reviewer_o54c · 2025-10-28

**Soundness:** 4
**Presentation:** 4
**Contribution:** 3
**Rating:** 6
**Confidence:** 3

**Summary:**

This paper tries to explain VLMs failures on simple visual tasks. They argue that this is due to failures in visually grounded serial processing. The authors compare VLMs and humans on tasks where serial processing requirements are manipulated. They find that VLM accuracy is correlated with human reaction times, which they take as a proxy for serial processing load. The authors here accumulate evidence for a hypothesis outlined in previous work [1]. Overall, I find the paper to be very well put together and enjoyable, and while it does not offer an entirely novel account of why VLM visual processing fails (see [1]), it does provide more evidence for this account and therefore I am inclined to accept it.

**Strengths:**

- *Originality* The paper is well motivated and uses established experiments to investigate a theory explaining problems in VLM visual processing.
- *Quality* The paper is very well put together, the experiments are sensible and overall it is a good read.
- *Clarity* The paper is written clearly and the results are presented in a very digestible way.
- *Significance* The paper accumulates further evidence for the binding problem hypothesis explaining the problems VLMs have with basic visual processing.

**Weaknesses:**

While I really like the paper in general, it does not offer an entirely novel account of why VLM visual processing fails (see [1]). Ideally, this paper would offer a more concrete path towards implementing visual serial processing in vision language models --  I think the different solutions offered in this paper are not worked out well enough to really understand what would fix this issue. In any case, I think the paper still provides novel evidence for the binding problem theory, albeit limited in scope somewhat, and I will recommend acceptance.

**Questions:**

**Main questions**
- Line 440 "However, this seems to be a strictly linguistic form of serialization that is difficult to apply to visual tasks like mental rotation" can you elaborate on this? Does this mean that since reasoning models can only generate new text they can not apply serialized processing to the visual information? I don't think I understand completely.
- Can you elaborate a bit on how exactly you perform the tool use experiments? I couldn't find a lot of detail on how this was run.
- One thing that would be nice to add is some data on what happens when you restrict exposure times or reaction times in humans. In theory, if serial processing is the only distinguishing thing between humans and VLMs in these tasks, if you restrict the time humans can engage with a specific stimulus shouldn't you see a similar drop in accuracy as in VLMs?

**Minor comments**
- I would put Figure 1 b-d on the same y-axis, the axes are close enough already and that would make it more easily comparable.
- Figure 2 the b for subfigure b seems to be cut off and the legends are not sorted in a coherent way (extremely minor).


[1] Campbell, Declan, et al. "Understanding the limits of vision language models through the lens of the binding problem." Advances in Neural Information Processing Systems 37 (2024): 113436-113460.

---

> ### Author Response · Authors · 2025-11-22
>
> We thank the reviewer for their positive assessment and insightful comments about our work.
>
> ## Novelty
>
> We agree that Campbell et al. established a critical baseline by identifying the "binding problem" as a key bottleneck. Our work builds upon this foundation but offers a distinct and necessary advancement. We have revised our manuscript (specifically our Section 8: Discussion) to clarify that our work moves beyond the general assertion that VLMs struggle with binding.
>
> By systematically testing VLMs on tasks that isolate different strategies (geometric program induction, perceptual enumeration, mental rotation), we seek to identify precisely which human visual reasoning capabilities VLMs lack and why.
>
> Concretely, we move beyond general binding failures to demonstrate:
>
> 1. **Scope**: The deficit is not limited to static binding; it extends to dynamic transformations (Mental Rotation) and compositional abstraction (Geometric Reasoning).
> 2. **Mechanism**: By systematically testing VLMs on tasks that isolate different human strategies (e.g., subitizing vs. counting, static vs. rotating), we show that VLMs rely heavily on linguistic scaffolding. Our results in the colored numerosity task prove that VLMs fail specifically when they cannot use language (e.g., linguistic referents to specific objects in the scene) to substitute for the visual serial attention they lack.
> 3. **Quantification**: We introduce human Reaction Time (RT) as a continuous, parametric predictor of VLM failure, providing a quantitative link that was previously missing.
> 4. **Causality**: we causally test VLM-specific reasoning strategies to determine whether augmentations (linguistic grounding via CoT, tool use, etc.) can compensate for these deficits.
>
> We have clarified these contributions in the paper.
>
> ## Concrete path towards solution
>
> We have revised our discussion to highlight promising directions: Visually Grounded RL (learning to "saccade" sequentially) and Recurrent Vision Architectures (iterative state updates). Our new attention analysis (Appendix E; see response to Reviewer Ddut) shows VLM attention becomes diffuse in overlapping conditions, confirming architectural failure in object individuation.
>
> ## Clarification on Line 440
>
> Correct. We distinguish linguistic serialization (generating sequential text tokens) from visual serialization (applying sequential operations to a visual representations), now clarified in Section 8.
>
> Specifically, our results show that the model’s linguistic serialization (Chain-of-Thought) fails when the task steps cannot be easily grounded in language:
> 1. **Mental Rotation**: As noted in Section 6, rotation involves continuous, analog transformations. These gradual spatial updates are difficult to discretize into a sequence of text tokens, causing the linguistic serial process to fail where the human visual process succeeds.
> 2. **Numerical Estimation**: As shown in Section 5, VLM serialization works only when objects can be linguistically referenced (e.g., by unique colors). When objects are identical and overlapping, the model cannot rely on language to serialize them (it cannot 'name' them distinctively), whereas humans can still serially individuate them.
> 3. **Geometric Reasoning**: As discussed in Section 4, human difficulty scales with program complexity (MDL), reflecting a step-by-step deconstruction of the shape. The VLM fails to replicate this compositional scaling, suggesting it cannot effectively serialize the visual primitives without explicit linguistic cues.
>
> ## Tool use
>
> Please see our response to Reviewer GX7P for full details (Appendix D.2, Table 2).
>
> ## Restricting RT in humans
>
> The reviewer posits that if the serial processing hypothesis is correct, restricting human time should force a collapse in performance similar to that of VLMs. This is a great suggestion. We would predict that under these conditions, human performance would collapse to match VLM performance (e.g., failing at rotation, failing at high numerosity).
>
>
> While running a new human psychophysics study is outside the scope of this rebuttal period, we note that established literature confirms this prediction:
> - Mandler & Shebo (1982) demonstrated that when exposure time is restricted (<200ms), humans can only accurately process small sets (subitizing range) and fail at larger exact counts, mirroring the VLM performance profile we observe.
> - Hertzog et al. (1993) showed that enforcing response deadlines causes human accuracy to plummet specifically for larger rotation angles, as the serial analog simulation is truncated.
> - Treisman & Gelade (1980) showed that brief exposures prevent the serial binding of features (e.g., in conjunction search), forcing humans to rely on parallel “pop-out" features.
>
> We have added a new paragraph to the **Discussion (Section 8)** citing established psychophysics literature that confirms this prediction.

---

> > ### Comment · Reviewer_o54c · 2025-11-27
> > **Reviewer response to rebuttal**
> >
> > Dear authors,
> >
> > I appreciate the detailed response, thank you! I enjoyed reading this paper and see the concerns I raised in my initial review as addressed, I am therefore raising my score.

---

### Official Review · Reviewer_Ddut · 2025-10-29

**Soundness:** 3
**Presentation:** 4
**Contribution:** 3
**Rating:** 6
**Confidence:** 4

**Summary:**

This paper addresses a key paradox in Vision-Language Models (VLMs): strong performance on complex benchmarks yet frequent failures on simple visual reasoning tasks. The authors propose the serial processing deficit hypothesis, arguing that VLMs lack visually grounded serial processing capabilities that humans rely on for tasks like geometric reasoning, enumeration, and mental rotation. Using controlled cognitive science paradigms, they evaluate four state-of-the-art VLMs (GPT-4o, Claude Sonnet 3.7, Llama 4 Maverick, Qwen2.5 VL) alongside human behavioral data. Human reaction time (RT) serves as a proxy for serial processing load. Across all domains, VLM accuracy inversely correlates with human RT, supporting the hypothesis. Augmentation strategies (Chain-of-Thought, tool use, reasoning training) improve performance but reveal limitations, suggesting the need for architectural changes to integrate serial visual reasoning.

**Strengths:**

- The serial processing deficit hypothesis offers a coherent lens for interpreting diverse VLM failures (counting, spatial reasoning, compositionality).

- Experiments are grounded in validated cognitive paradigms with parametric control over complexity (MDL for geometry, overlap/color for enumeration, angular disparity for rotation).

- Using human reaction time (RT) as a continuous predictor of VLM performance bridges cognitive science and AI evaluation, enabling predictive insights beyond static benchmarks.

- The paper is well-written, with relevant figures and comprehensive appendices detailing prompts, stimuli, and per-model results.

- Discussion of architectural directions (visually grounded RL, foveal masking) provides a roadmap for addressing identified limitations.

**Weaknesses:**

- “Serial processing” conflates multiple mechanisms (attentional shifts, compositional reasoning, iterative refinement). It is good to clarify which aspects are hypothesized to be deficient.

- RT is used to define and validate serial processing load, but RT reflects overall task difficulty, not exclusively serial processing. Alternative explanations (binding problem, perceptual ambiguity) are not fully ruled out.

- Augmentation strategies (CoT, tool use) provide task-specific scaffolding rather than true serial visual processing. Improvements in colored overlapping conditions suggest linguistic grounding, not visual seriality.

- The paper treats VLMs as black boxes without probing attention patterns or internal representations. Linking performance to architectural features would strengthen conclusions.

- While architectural improvements are discussed, none are implemented or evaluated.

- Findings are based on synthetic stimuli and four VLMs; applicability to real-world images and newer architectures remains uncertain.

**Questions:**

Check weakness in addition to:

- How do you dissociate serial processing from general task difficulty in the RT–accuracy relationship?

- How is the serial processing deficit distinct from the binding problem? Could these be alternative framings of the same phenomenon?

- Have you analyzed VLM attention maps for evidence of sequential scanning?

- Could failures in mental rotation reflect insufficient training on rotated exemplars rather than architectural limitations?

---

> ### Author Response · Authors · 2025-11-22
>
> Thank you for your constructive feedback. We address specific questions below.
> ## Serial processing mechanisms/binding problem
>
> We clarify the definition of serial processing (and how it relates to the binding problem) below. We added them to the revised paper.
> - We define “visual serial processing" based on the cognitive science literature (Treisman & Gelade, 1980; Ullman, 1984) as the iterative application of operations (spatial indexing, attention shifting, analog transformation) to a visual representation. The specific operation varies by domain, but the bottleneck is the same: the need to maintain/update a visual state over time.
> - We view the serial processing deficit as the mechanistic cause of the binding problem. Humans solve the binding problem by serially attending to individual objects. When this mechanism fails, binding fails.
> ## RT v. Difficulty
>
> As the reviewer notes, RT reflects difficulty. However, in human vision, the mechanism for handling increased difficulty in these domains is serial processing. It is not just that the tasks are hard, but that human RT scales **linearly** with the parameter of interest (e.g., number of items, rotation angle). In cognitive science, this linear slope is the signature of serial processing (Shepard & Metzler, 1971).
>
> We hypothesize that VLMs lack this serial mechanism (or are limited in their serialization), so they cannot adapt to reasoning difficulty (other work identifies a correspondence between human and LLM reasoning effort in text-only domains, see Varda et al. 25). Consequently, we see that precisely where humans show the linear signature of serial adaptation (high RT), VLM performance collapses.
>
> We have added a paragraph to Section 3 (The serial processing deficit hypothesis) justifying our use of RT linearity as a specific proxy for serial load.
>
> ## Task-specific scaffolding in augmentation
>
> We are in agreement. In fact, this is a central finding of our paper.
> - Augmentation as task-specific scaffolding: We draw this conclusion in section 7. Our goal is not to claim they solve the problem, but to diagnose it. We state: “These results highlight the limitations of existing methods… as their application is limited to specific settings.”
> - Colored overlapping results suggest linguistic grounding: We predict this result in Section 7, and conclude in Section 8 that “this advantage… arises because distinct colors allow VLMs to… linguistically ground features… in a way that is not possible in uniformly colored conditions.”
>
> We now explicitly frame these experiments as diagnostic analyses rather than proposed solutions.
>
> ## Probing Internal Representations (Attention Analysis)
>
> We conducted a new analysis of attention mechanisms in Qwen2.5-VL-7B and LLaVA-v1.6-Mistral-7b on the numerosity task. The results of this analysis have been added to the new Appendix E (referenced in Section 8.1).
>
> Method: We generated attention maps over the image tokens for each text token, then evaluated the correlations in object-level attention over the objects in these conditions. (See Appendix for more details). A low correlation indicates that the model is attending to objects independently, a hallmark of object-centric processing, while a high correlation suggests that the model treats the objects as a single group rather than as distinct entities.
>
> We found strong evidence that the VLM failure stems from a **failure in object individuation** during visual reasoning:
> - We find higher attention correlations in the overlapping condition compared to the non-overlapping condition (Qwen: $t=14.06, p<0.001$; LLaVA: $t=2.74, p=0.006$). This indicates the deficit stems from a failure to individuate overlapping objects while reasoning. When objects overlap, the model’s attention becomes more diffuse and less object-selective.
> - In LLaVA-v1.6, we observed a significant decrease in attention correlation in the colored condition compared to uniform conditions ($0.799$ vs $0.888$, $t = -18.57$, $p < 0.001$), suggesting color enables more object-centric attention even with spatial overlap.
> These strongly align with our behavioral results.
>
> ## Rotation training
>
> We believe this is unlikely to be the primary factor for two reasons:
> 1. Modern vision backbones are trained with massive data augmentation, including random rotations.
> 2. If it were merely a data distribution issue, we would expect smoother degradation or better generalization for moderate angles. Instead we see a sharp drop-off that mirrors degradation in human performance under time pressure.
>
> ## Synthetic Stimuli
>
> We chose synthetic stimuli to allow for fine-grained control over serial processing load (MDL, rotation angle, overlap) while controlling for other factors (e.g., context clues) that mask the deficit in real world images. VLM performance deficits in naturalistic visual reasoning tasks are well-documented in the literature (see Section 2.2). Our focus is to isolate the specific causes of these deficits.

---

> > ### Comment · Reviewer_Ddut · 2025-11-25
> >
> > I thank the authors for their responses. I have few followup questions based on overall analysis and reviews submitted.
> > - Human RT increases with task complexity, but does it uniquely reflect serial processing? Could it equally reflect perceptual difficulty, stimulus ambiguity, or decision uncertainty?
> > - Appendix E shows higher attention correlations under overlap, but does this prove object individuation failure or just diffuse attention?
> > - Could failures in Enumeration reflect insufficient visual working memory (capacity) rather than lack of sequential attention? However, I note that Rotation failures (N=2) argue against this; do you view the 'bottleneck' as a singular deficit across these domains, or are VWM limits and serial processing deficits distinct failures for VLMs?
> > - Three tasks chosen are specifically non-linguistic (geometric, enumeration, rotation), but VLMs excel at linguistic tasks (VQA, captioning). Do findings generalize to visual reasoning tasks where language can scaffold (e.g., "what is the relationship between these objects")?

---

> > > ### Author Response · Authors · 2025-11-26
> > > **Thank you for your continued engagement. We address each point below.**
> > >
> > > **Human RT:**
> > > In human vision, the mechanism for handling increased task difficulty is serial processing. Therefore, “task difficulty” and “serial processing” are not competing hypotheses. Rather, **serial processing is a strategy humans use to adapt to task difficulty.** The difficulties the reviewer suggests may be handled via serial processing in some domains. For example, serial processing in our numerosity task involves operations meant to disentangle perceptual difficulty and stimulus ambiguity. In the paradigms we test, a linear increase in RT is a well-established, standard behavioral signature of serial processing (Shepard & Metzler, 1971). See our revised Section 3 for a full justification.
> > >
> > > **Appendix E:**
> > > Object individuation failure and diffuse attention are likewise not competing hypotheses. Diffuse attention is a signature of a failure to individuate objects. Our correlation metric captures synchronization of object-wise attention time series, i.e., grouped/diffuse attention across objects. We interpret higher correlations under overlap as evidence that the model is treating the objects as a single group rather than as distinct entities. Taken together with (i) decreasing accuracy in overlapping conditions and (ii) increasing accuracy and attention decorrelation when objects are colored, these findings provide converging support for a serial-processing deficit. This can be validated by inspecting example CoTs from each condition:
> > >
> > > *Uniform, non-overlapping condition; n=8; trial # 68; Claude-Opus:*
> > > > “Looking at the image, I'll count each object systematically:
> > > > 1. Top left: One triangular shape
> > > > 2. Left side, upper: One pentagon/hexagon shape
> > > > 3. Left side, middle: One pentagon/hexagon shape
> > > > 4. Left side, lower: One pentagon/hexagon shape
> > > > 5. Bottom center: One irregular polygon (flower-like shape)
> > > > 6. Top center-right: One irregular polygon (cloud-like shape)
> > > > 7. Right side: One triangular shape
> > > > 8. Right side, lower: One small quadrilateral shape
> > > > All objects appear to be separate and non-overlapping simple outlined shapes.”
> > >
> > > *Uniform, overlapping condition, n=8, Claude Opus:*
> > >
> > > **Trial # 3:**
> > > > “I can see several overlapping abstract shapes drawn with black lines. Let me identify each object carefully:
> > > > 1. On the left side, there's an irregular curved shape that forms a closed loop
> > > > 2. In the upper portion, there's another curved/organic shape that overlaps with others
> > > > 3. There's a triangular or angular shape visible in the lower portion
> > > > 4. In the center-right area, there's another irregular closed shape
> > > > 5. There appears to be another curved shape on the right side
> > > > 6. I can identify what looks like another angular/triangular shape in the central area
> > > > The objects are heavily overlapping in the center, making it challenging to distinguish exactly where one ends and another begins. After carefully tracing each complete closed shape, I count 6 distinct objects in total.”
> > >
> > > **Trial # 31:**
> > > > “I can see overlapping geometric shapes with black outlines. Let me identify each object:
> > > >
> > > > On the left side, there's a quadrilateral shape (appears to be a tilted rectangle or parallelogram)
> > > >
> > > > In the upper-right portion, there's a pentagon or irregular polygon shape
> > > >
> > > > In the lower portion, there's another polygon shape (appears to be an irregular hexagon or heptagon)
> > > >
> > > > These three shapes are overlapping each other, creating a complex arrangement where parts of each shape are visible while other parts are hidden behind the overlapping shapes.”
> > >
> > > **Visual working memory:**
> > > We agree that a pure VWM-capacity account is unlikely to explain our data. In numerosity, set size is matched across conditions, yet performance selectively collapses for overlapping displays and is rescued when objects are color-marked; color does not reduce the number of items to be stored, but instead facilitates segmentation and indexing, which points to a deficit in object individuation/serial access rather than “too few slots.” Consistent with this, rotation failures already appear at N=2, where a capacity explanation is implausible. We therefore view the pattern across tasks as reflecting a shared bottleneck in forming and sequentially accessing object-indexed representations, with apparent VWM limits in these VLMs emerging from this serial-processing deficit rather than from a distinct, human-like capacity limit.
> > >
> > > **Linguistic scaffolding (VQA):**
> > > We would expect tasks in which language can scaffold reasoning to be less challenging for models. That said, when the relations are specifically spatial relations among objects, this may challenge models as they require more object centric attention. These are the kinds of relational reasoning tasks that challenge existing models (Liu et al. 2023) and that are the focus of recent efforts to address spatial reasoning deficits in VLMs (e.g., Spatial VLM, Chen 24). These datasets specifically construct reasoning chains that linguistically ground spatial reasoning.

---

### Official Review · Reviewer_GX7P · 2025-11-04

**Soundness:** 3
**Presentation:** 3
**Contribution:** 3
**Rating:** 4
**Confidence:** 4

**Summary:**

This paper proposes the "serial processing deficit hypothesis" to explain systematic failures of VLM on visual reasoning tasks. The authors test this hypothesis across three cognitive domains (geometric reasoning, numerical estimation, and mental rotation) by systematically varying task complexity. The core finding is a consistent inverse correlation between human reaction time (used as a proxy for serial processing load) and VLM accuracy: as tasks require more sequential processing (indicated by longer human RT), VLM performance degrades significantly while human accuracy remains high. The paper provides causal evidence through augmentation experiments showing that external serial processing mechanisms (CoT, reasoning training, tool use) can improve performance in limited contexts, but VLMs lack intrinsic visual serial reasoning capabilities.

**Strengths:**

1. Using human reaction time as a proxy for serial processing load is well-grounded in cognitive science and provides a principled way to manipulate task complexity across diverse domains.

2. Testing across three distinct cognitive domains (geometric reasoning, numerosity, mental rotation) with multiple SOTA VLMs (GPT-4o, Claude Sonnet 3.7, Llama 4 Maverick, Qwen2.5 VL) strengthens the generalizability of findings.

3. The observation that CoT only helps counting when objects have unique colors reveals the fundamental limitation—VLMs can only leverage serial processing when it can be linguistically mediated.

**Weaknesses:**

1. **Missing critical prior work on VLM serial processing limitations**: This problem has been substantially investigated in prior work, particularly in "How Far Are We from Intelligent Visual Deductive Reasoning?" (arXiv 2403.04732, March 2024). That paper:
   - Documented that VLMs fail at visual deductive reasoning tasks requiring multi-hop comparative reasoning and sequential hypothesis verification
   - Showed that **segmenting RPM problems into individual patterns improved accuracy from 7.4% to 21.2%**—direct causal evidence that breaking tasks into sequential steps helps VLMs
   - Identified "compounding errors" where VLMs make cascading mistakes in autoregressive pattern description, suggesting inability to perform iterative self-correction
   - Demonstrated VLMs achieve only 10-20% accuracy even when in-context examples are identical to queries, showing failure in sequential visual comparison
   - Found that VLMs make "confounding errors" where they cannot maintain focus on single patterns when multiple similar patterns are present—a clear attention/serial processing deficit

   The current paper claims the serial processing deficit as a key novel finding but fails to cite or discuss this highly relevant prior work that provides converging evidence from a different task domain (abstract geometric pattern reasoning). The segmentation results in particular are compelling causal evidence that the authors should engage with.

2. **Limited architectural analysis**: While the paper identifies the deficit, it provides minimal insight into *why* current VLM architectures fail at serial processing. Is it the single-pass vision encoder? Lack of recurrent processing? Insufficient visual working memory?

3. **Weak tool use baseline**: The tool use augmentation only tests rotation tools for the mental rotation task. Why not test cropping/zooming tools for counting tasks, or tools that isolate geometric elements? The current tool use experiments don't adequately test whether VLMs can leverage external mechanisms for serial visual analysis beyond simple image transformations.

4. **Insufficient analysis of human-VLM differences**: The paper uses human RT as ground truth but doesn't deeply investigate *how* humans perform serial processing on these tasks. Eye-tracking data or think-aloud protocols could provide richer insights into what specific serial mechanisms are missing in VLMs.

**Questions:**

1. How do you reconcile your findings with the RPM paper's segmentation experiments, which provide direct causal evidence for the serial processing deficit?


2. Can you provide more granular analysis of where in the processing pipeline VLMs fail? For example, do they fail at individuating objects (segmentation), tracking them (working memory), or transforming representations (mental operations)?

3. The CoT results suggest linguistic mediation is necessary for serial processing in current VLMs. Could multimodal chain-of-thought (with intermediate visual representations) help?

4. Have you tested with few-shot evaluation?

---

> ### Author Response · Authors · 2025-11-22
>
> We thank the reviewer for their detailed feedback and for identifying the missing citation. We address specific points below.
>
> ## Prior work (arXiv 2403.04732):
> We've added discussion of this paper to Section 2.2. It provides strong evidence for our hypothesis in discrete, deductive visual reasoning (RPMs), aligning with Campbell et al. (2024).
>
> We make three new contributions:
> 1. Grounding in multi-object reasoning tasks (Zhang et al., Campbell et al.) is a **distinct** form of multi-hop reasoning from the continuous transformations of visual representations necessary to solve the tasks in our work (e.g., mental rotation). Our tasks require a form of visual processing that is difficult to ground linguistically **irrespective of relational reasoning**.
>
> 2. Our numerosity task cleanly dissociates general reasoning difficulty from linguistic grounding difficulty. As we show, when models can ground salient features (like color) in language, performance improves, isolating the failure to a lack of visual (non-linguistic) serialization.
>
> 3. A core methodological contribution of our paper is the use of human reaction time (RT) as a domain-general, continuous measure of serial processing load. This allows us to show a quantitative, inverse correlation between human serial load and VLM accuracy across all three distinct domains.
>
>
> We have revised Section 2.2 to discuss Zhang et al. (2024) and frame our contributions in this context.
>
> ## Architectural analysis
> The reviewer asks for a "granular analysis of where in the processing pipeline VLMs fail," specifically regarding object individuation. To address this, we conducted a new analysis of the internal attention mechanisms of two open-source models (Qwen2.5-VL-7B and LLaVA-v1.6-Mistral-7b) on our numerosity task. We added these results to Appendix E, referenced them in Section 8.1, and you can find a further detailed explanation in the rebuttal response to reviewer Ddut.
>
>
> ## Tool Use
> As detailed in Appendix D.2, **our tool use experiments were run across all three domains**. We curated a challenging subset of trials from Geoclidean (MDL 3 & 4), Numerosity (n = 8), and Rotation (high angles). The full results are reported in **Table 2**. We have made this more clear in the revised paper.
>
> The experiments used GPT-o3 with advanced agentic visual analysis tools (cropping/zooming, object extraction, geometric element isolation, code execution, and other fine-grained operations beyond simple rotation).
>
> **The results from the experiment strongly support our central hypothesis**. As shown in Table 2, when given these advanced visual tools, the model achieves near-perfect accuracy on rotation and non-overlapping counting, but chance performance on the overlapping conditions (which cannot be solved by ‘zooming in’). This finding is a key piece of causal evidence that these powerful agentic tools are still not a substitute for the visually-grounded serial processing that humans employ. We have updated **Appendix D.2** to clarify our approach and results.
>
> ## Mechanisms for human serialization
> The reviewer suggests using eye-tracking or think-aloud protocols. While we agree these are valuable methods, these mechanisms are already well-documented in the cognitive science literature motivating our work:
> - **Mental Rotation**: Decades of eye-tracking research (Just & Carpenter, 1976) confirm that human eye movements linearly track the rotation path, proving the mechanism is an analog serial transformation.
> - **Numerosity**: The distinction between parallel subitizing and serial counting is foundational (Trick & Pylyshyn, 1994). The mechanism for counting large sets is known to be serial spatial attention (Treisman & Gelade, 1980).
>
> We are not using reaction time (RT) solely as a measure of difficulty. In these specific paradigms, a **linear increase** in RT is the standard behavioral signature of these serial mechanisms (Shepard & Metzler, 1971). We have added a paragraph to Section 3 explicitly referencing this literature to justify our use of RT as a proxy for these specific serial operations.
>
> ## Multimodal CoT and Few-shot evaluation
> - **Multimodal CoT**: We agree this is a promising direction. However, as our new attention analysis indicates, the core deficit arises due to the failure of the multimodal decoder to effectively individuate objects. If the model cannot visually segment objects, generating intermediate reasoning steps is unlikely to succeed. We view this as an architectural challenge for future work, and have highlighted it in the Future Work section of the revised paper.
>
> - **Few-Shot**: As noted by Zhang et al. (2024), standard few-shot strategies often yield limited gains in visual reasoning tasks. Our focus in this work was on evaluating the intrinsic zero-shot serial processing capabilities of the models to isolate the architectural deficit. We have noted this and clarified the motivation for our approach in the revised paper.

---

### Author Response · Authors · 2025-12-04

To aid in the final decision, we summarize our work and updates made following the review discussion.

## Summary of Contributions
Our paper proposes the “Serial Processing Deficit Hypothesis” to explain why VLMs fail on seemingly simple visual tasks such as counting and mental rotation despite strong benchmark performance. We show that human reaction time (RT), a proxy for serial processing demand, is a strong inverse predictor of VLM accuracy across three distinct cognitive domains.

## Novelty and Contributions
While recent benchmarks have documented *when* VLM performance falters, they lack a unifying explanation for *why*. Previous work has identified specific bottlenecks: the binding problem and deficits in multi-hop reasoning.
Our work unifies these explanations by drawing on visual reasoning tasks from cognitive science. We highlight that human visual reasoning employs a varied repertoire of strategies beyond the serial spatial attention needed for binding or multi-object reasoning. By systematically testing VLMs on tasks that isolate different strategies (geometric program induction, perceptual enumeration, and mental rotation), we identify precisely which human visual reasoning capabilities VLMs lack and why their strategies fail.

We attribute VLM failures to over-reliance on linguistic grounding, showing they fail when non-linguistic strategies are required:

**Geometric Reasoning**: We replicate findings that human RT scales with structural complexity (MDL). VLM accuracy shows a strong inverse correlation with human RT, suggesting sensitivity to abstract structural complexity but lacking the serial mechanisms to maintain accuracy as complexity increases.

**Perceptual Enumeration**: Under high object overlap requiring effortful serial attention in humans, VLM performance degrades significantly. Crucially, uniquely coloring overlapping objects substantially improves model performance, a benefit not seen in humans, suggesting that VLMs lack human-like flexible visual attention and rely instead on language to scaffold reasoning.

**Mental Rotation**: Human RT scales linearly with rotation angle, but VLM accuracy collapses to chance beyond ~45°. This demonstrates that models lack the dynamic strategies required to mentally rotate objects.

In summary, our work seeks to isolate failure modes across a wider range of visual reasoning strategies than previously explored. We provide causal evidence that current VLM reasoning is overly reliant on linguistic grounding (CoT & reasoning) and simple 2D image transformations (image tool use), failing to capture the richer, internal visual transformations and attentional strategies characteristic of human cognition.

## Reviewer Response and Updates
Three reviewers (GX7P, Ddut, o54c) responded positively, highlighting the “excellent” presentation, “well-grounded” use of human RT, and strength of testing across diverse domains. We conducted additional experiments during discussion to address specific concerns:

1. **Architectural Analysis (Reviewers GX7P, Ddut)**: We conducted attention analysis on open-source models (Qwen2.5-VL, LLaVA-v1.6), now in Appendix E.
_Result_: In conditions requiring serial attention, VLM attention becomes diffuse and highly correlated across objects, providing mechanistic evidence that performance collapse stems from failure to individuate objects during encoding.

2. **Tool Use and Augmentation (Reviewers GX7P, o54c)**: We clarified our evaluation of models capable of manipulating images when reasoning (cropping, zooming, etc.,) across all three domains. Tools-use improved performance on the mental rotation and spatially separated counting tasks but failed at overlapping counting, reinforcing our hypothesis.

3. **Novelty and Prior Work (Reviewers GX7P, o54c)**: We revised text to clarify distinctions from prior work (Campbell et al., 2024; Zhang et al., 2024) and included missing citations. Following these clarifications, Reviewer o54c raised their score, noting their concerns were addressed.

4. **Validity of Reaction Time (Reviewers Ddut, kB3k)**: We clarified (Section 3) that we use RT specifically as the linear RT slope, the established behavioral signature of serial processing in cognitive science (Shepard & Metzler, 1971), not as a generic difficulty measure.

We thank the reviewers for their constructive feedback, which prompted new experiments that have significantly strengthened our work.​​​​​​​​​​​​​​​​

---

### Note · Program_Chairs · 2026-01-17
**Submission Desk Rejected by Program Chairs**

The following references in this submission do not refer to real documents and/or have major errors in bibliographic information:

 Kevin Sablé-Meyer, Adam Santoro, Brenden M Lake, Joshua B Tenenbaum, Jean-Louis Dessalles, Steven T Piantadosi, Radoslaw Martin Cichy, and Stanislas Dehaene. Language as a bottleneck: a cross-cultural study of the efficient coding of geometric shapes. Proceedings of the National Academy of Sciences, 119(26):e2123203119, 2022.